# Multidomain Joint Learning of Pedestrian Detection for Application to Quadrotors

**Yuan-Kai Wang** [1] , **Jonathan Guo** [1] **and Tung-Ming Pan** [2,*]

1   Department of Electrical Engineering, Fu Jen Catholic University, New Taipei 242, Taiwan
2   Holistic Education Center, Department of Information Innovation and Digital Life, Fu Jen Catholic University, New Taipei 242, Taiwan
*   Correspondence: macgyver@fju.edu.tw

**Abstract:** Pedestrian detection and tracking are critical functions in the application of computer vision for autonomous driving in terms of accident avoidance and safety. Extending the application to drones expands the monitoring space from 2D to 3D but complicates the task. Images captured from various angles pose a great challenge for pedestrian detection, because image features from different angles tremendously vary and the detection performance of deep neural networks deteriorates. In this paper, this multiple-angle issue is treated as a multiple-domain problem, and a novel multidomain joint learning (MDJL) method is proposed to train a deep neural network using drone data from multiple domains. Domain-guided dropout, a critical mechanism in MDJL, is developed to self-organize domain-specific features according to neuron impact scores. After training and fine-tuning the network, the accuracy of the obtained model improved in all the domains. In addition, we also combined the MDJL with Markov decision-process trackers to create a multiobject tracking system for flying drones. Experiments are conducted on many benchmarks, and the proposed method is compared with several state-of-the-art methods. Experimental results show that the MDJL effectively tackles many scenarios and significantly improves tracking performance.

**Keywords:** pedestrian detection; multiobject tracking; multitask learning; multidomain joint learning; MDJL; drone application

## 1. Introduction

Pedestrian detection and tracking applications involving intelligent robots or flying drones [1–3] facilitate the determination of the position of a human to avoid collisions and the appropriate direction of a camera when walking or acquiring images. Both applications, which are used in autonomous cars, can assist computers in determining whether braking should be performed to slow down or avoid pedestrians [4,5].

Pedestrian detection and tracking in flying drones are used as examples because these miniature aircraft can move arbitrarily in the sky; the acquired images will be different because of the flight height and angle of the mounted camera. Thus, pedestrian images obtained with a downward orientation are different from those acquired with a forward-pointed camera. This causes significant challenges when pedestrian detection and tracking are applied to drones.

However, to train a network that covers multiple domains in an application scenario, the collection of videos and the preparation of annotations to create a dataset that meets our needs is time-consuming and labor intensive. Such multiple domains can then be integrated to use the resulting multidomain datasets [6]. However, after referring to the relevant literature regarding this approach, we found that such datasets will influence each other in each domain, making it difficult for the network to learn how to extract key features, reducing performance. This paper proposes a training method based on domain-guided dropout [7] that uses network domain information to enhance the performance of

the network when learning. In addition, our detection method was applied to a pedestrian tracking system that is suitable for drones.

In practice, it is time-consuming to collect video data and generate the corresponding annotations. The most intuitive method is to combine multiple ready-made datasets to train the model. However, in the experiments, the classification accuracy of the model trained with this method was slightly reduced when applied to the test data of each dataset. We believe that this phenomenon occurred because the different video datasets have different scenes, angles, shooting lengths, and equipment. Therefore, the features of the video data in each dataset varied. The neural network is easily affected by these differences during training, and learning a method for efficiently extracting pedestrian features is impossible, resulting in performance degradation.

In computer vision, a domain can be broadly referred to as a collection of data groups with the same features. For example, Figure 1 shows three pedestrian detection datasets that belong to different domains. The objective is to train the pedestrian detection network by combining the datasets of multiple domains so that the learned model can be more generalized to cope with the challenges of different domains.

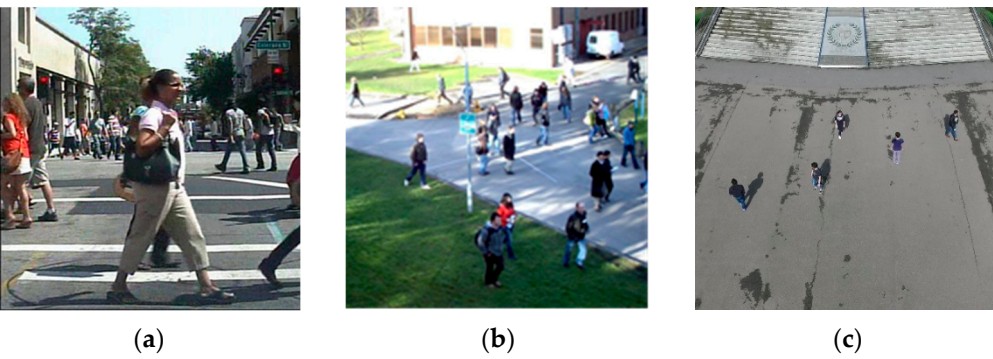

(**a**)              (**b**)              (**c**)

**Figure 1.** Pedestrian detection datasets belonging to different domains: (**a**) videos recorded using the driving recorder, (**b**) videos of campus surveillance, and (**c**) aerial videos obtained using flying drones.

The emergence of superlarge datasets, such as ImageNet, has led to the flourishing of neural networks [8]. Starting from the pretrained model using a superlarge dataset to perform further fine-tuned training, also known as transfer learning, has a better effect than using a randomly generated initial network. However, it is difficult to obtain this large dataset in various domains. Therefore, research units in different domains often create smaller datasets, but the simultaneous use of multiple datasets to train a model is challenging. In this paper, we call this dataset, composed of multiple datasets of different domains, a multidomain dataset.

Domain-guided dropout (DGD) [7] is a dropout method wherein neurons that respond strongly to a specific domain may not respond as strongly to other domains. By analyzing the degree of response of each neuron in the network to each domain, the probability of neuron dropout during the training can be adjusted. This improves the specificity and sensitivity of the neuron to features, thereby improving accuracy.

Our proposed pedestrian detection method, multidomain joint learning (MDJL), is derived from multitask learning. It uses the DGD characteristics to allow the network to effectively learn the features during training with the multidomain dataset to improve accuracy and generalizability across the domains. It was applied to the multi-pedestrian tracking algorithm for use in the changeable environment of a flying drone.

Figure 2 shows the flowchart for the proposed method. First, initialization training is performed so that the network can initially learn the features of each domain. The neuron impact score (NIS) of each domain is then calculated. The dropout in the network is replaced with the DGD, and training is continued. Finally, domain-specific fine-tuning is performed for each domain-specific neuron.

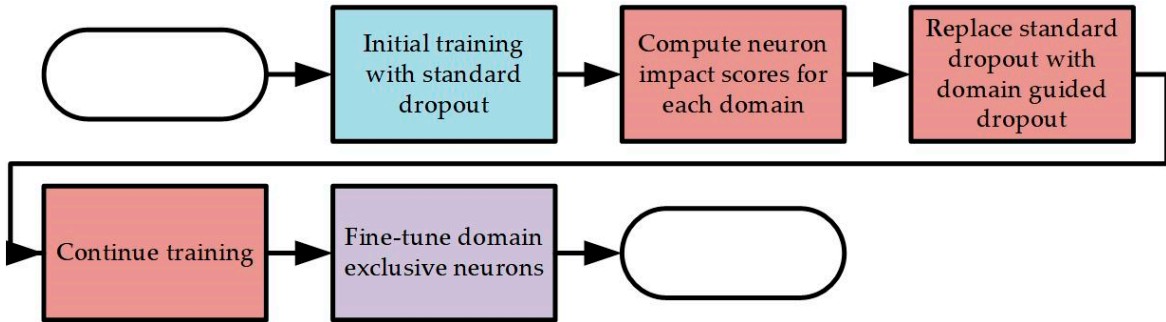

**Figure 2.** MDJL flow chart.

This process divides the network into several subnets, as shown in Figure 3, and each subnet is specialized to a specific domain. Given the characteristics of the DGD, subnets may overlap, which means that the domains have common data features.

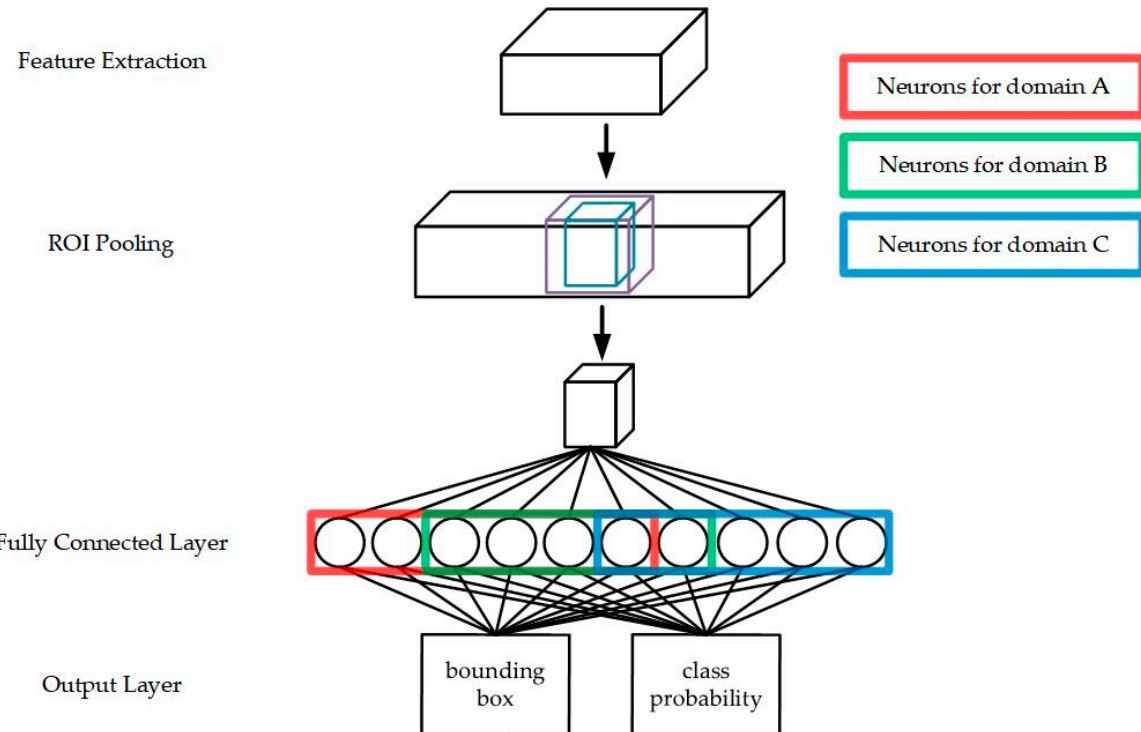

**Figure 3.** Schematic of MDJL subnet segmentation.

The remainder of this paper is organized into different sections. Section 2 reviews related deep learning methods and tracking schemes. The MDJL method is presented in Section 3. Experimental results and comparisons of the proposed technique with state-of-the-art methods are presented in Section 4. Finally, the conclusions and suggestions for future work are summarized in Section 5.

## 2. Related Works

Pedestrian detection and human pose estimation are typical surveillance applications [9–11]. Pedestrian detection algorithms can use traditional machine-learning methods on the basis of feature extraction and a learnable classifier. The architecture of this method extracts the specified features of the image (e.g., Haar feature [12], histogram of oriented gradients (HOG) [13], and deformable part models (DPM) [14]) and then uses sliding windows of different sizes to sample the feature images. These samples are finally sent to

a trained classifier (e.g., support vector machine (SVM) [15]) for classification to achieve pedestrian detection.

Traditional object detection methods typically use image pyramid and sliding-window methods to address a problem in which the size of objects in an image is not fixed. In addition, to reduce the number of samples for a single image, some methods analyze additional information extracted from the image, such as the dominant ground plane, to limit the sliding-window range [16]. Methods such as Viola–Jones [12], HOG [13], convolutional channel filter [17], and checkerboard [18] are the representative methods of traditional machine learning.

Another pedestrian detection algorithm is the neural-network method [19]. However, the developed multilayer perceptron (MLP) cannot efficiently process multidimensional video data. LeCun et al. [20] proposed the convolutional neural network (CNN) as network architecture. CNN can be divided into two layers: feature extraction and classification networks. Its function is equivalent to that of the HOG or DPM feature extraction methods. The features extracted by the CNN are not manually designed but are generated by the network via a learning stage. The classification network is equivalent to an SVM classifier, which is a set of classifications that learns how to use the information provided by a previous network. This neural network does not rely on manually designed feature extraction methods.

The advantage of CNN is that it can effectively solve manually designed feature extraction problems using traditional methods. Provided that the network parameters are adjusted via backpropagation using end-to-end training, the CNN can learn a filter to effectively extract the key features of the object and categorize the feature classifier extracted by this filter.

Some object detection methods using neural networks also use the sliding-window method [21] when the size of objects in an image is not fixed but the computation time is generally long. Therefore, in recent years, the mainstream method has been to adopt the strategy of object proposal by analyzing the bounding box that is likely to appear as the target object in the image. These possible bounding boxes are then inputted into the neural network for further analysis to obtain the final result. Moreover, some methods use the bounding box regression algorithm to correct the error of the bounding box for improved accuracy. In this study, algorithms such as faster R-CNN [22], Deepparts [23], UDN [24], and MS-CNN [25] were used as the baseline methods of deep learning.

MS-CNN uses different depths of CNN features with varying sizes of receptive fields to increase the detection range of object sizes. The implemented MS-CNN follows the feature-sharing concept proposed in [26] and uses a CNN trunk to detect long-distance object features. The difference is that the image features used by MS-CNN are extracted from different depths of the CNN trunk and sent to the region proposal network for analysis. This approach can address the problem in which the features of objects that are too small disappear in the deep CNN layer because of max pooling.

To enable the application of MS-CNN to drones, DGD, which is used in traditional neural-network training, is used to replace dropout [27]. Different datasets are regarded as independent domains, and the dropout probability is adjusted by analyzing the degree of response of each neuron to each domain. Consequently, the network can learn how to extract the most valuable features in each domain to improve its overall performance.

Recently, multiobject tracking methods have adopted the tracking-by-detection approach, which primarily aims to obtain the best connection between the bounding boxes in each frame. Therefore, most multiobject tracking methods aim to maximize a posteriori probability (MAP). The solutions to MAP problems can be divided into two categories. The first type of deterministic optimization problem is amenable to offline tracking methods. A commonly used method is based on a graphical solution, which uses the detection information to define a directed graph and obtain the best association tracklet via cost minimization [28,29]. Alternatively, an energy function can be utilized to detect the scene's

fixed action mode and the constraint rules. Energy minimization is then used to obtain the best solution [30].

The second method uses probabilistic inference and a probability inference method based on current detection information to estimate the target state distribution. Given that only present and past information is used, it is more suitable for online tracking methods. It uses single-object tracking to predict the location, surface feature, and motion to model the target. These two models are used to predict the location of a target object. The data association algorithm is then used to associate the relevant bounding box. Finally, the target model is updated, and the aforementioned steps are performed in a loop to achieve online multiobject tracking. The Kalman filter combined with the Hungarian algorithm for data association has been used to achieve multiobject tracking [31,32]. In addition to using the Kalman filter as the motion model, mean-shift tracking has been used as an appearance model, using the Hungarian algorithm to perform data association. The Markov decision-process (MDP) tracker [33] introduces the concept of learning to automatically obtain the similarity function for data association and uses the MDP model to determine the object's state in the adopted tracking strategy.

## 3. MDJL Method

Multitask learning (MTL) is a training method that has been commonly used in deep learning networks in recent years. The application of MTL includes natural language processing, speech recognition, and computer vision.

Figure 4 shows the differences between single- and multitask learning. Single-task learning is optimized for a specific task or even a particular domain during training and learning. However, MTL combines training information from multiple different but related tasks, such that the backpropagation results between different domains can compete to achieve a more generalized network model.

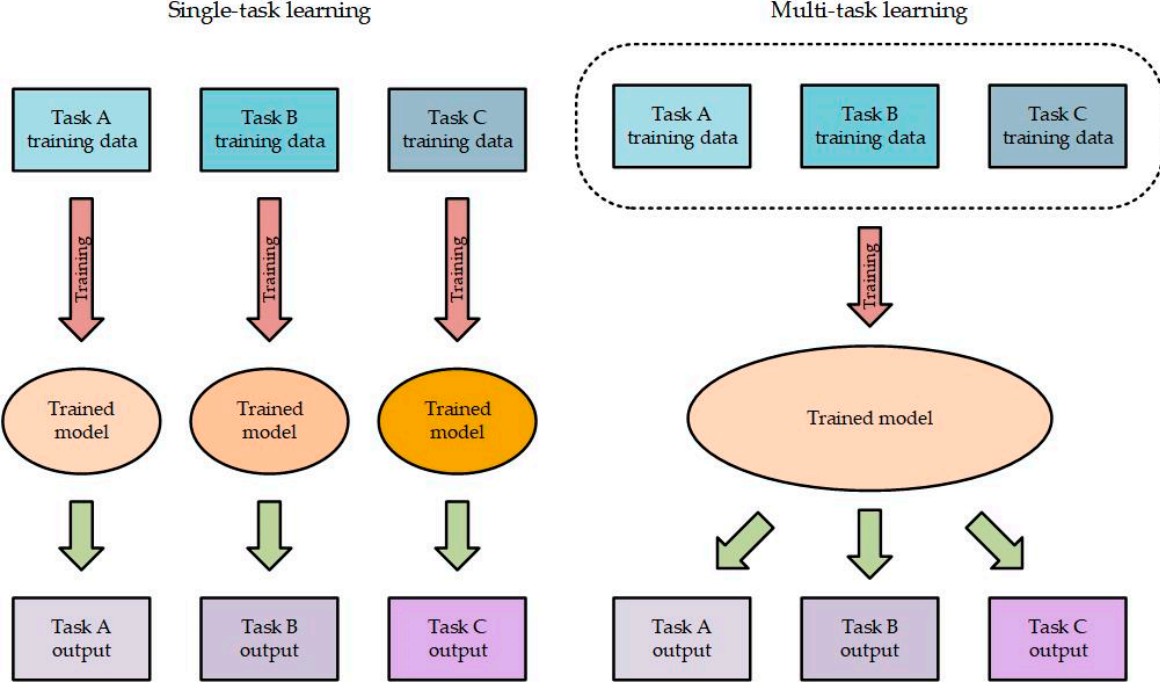

**Figure 4.** Single-task learning vs. multitask learning.

Labeled training data are available for a set of $k$ tasks $\mathcal{T} = \{\mathcal{T}_1, \mathcal{T}_2, \ldots, \mathcal{T}_k\}$, where each task is associated with a different domain, $\mathcal{D} = \{\mathcal{D}_1, \mathcal{D}_2, \ldots, \mathcal{D}_k\}$ [34]. It is nearly impossible to reliably estimate the empirical joint distribution $\hat{P}_k(X, Y)$ by using data from the kth domain only at the kth task, where $X = \{x_1, x_2, \ldots, x_n\}$ is the set of samples from the feature space and $Y = \{y_1, y_2, \ldots, y_n\}$ is the label of the sample. A better approximation

for $\hat{P}_k(X, Y)$ is learned by exploiting the training data from all domains and learning all tasks simultaneously. Thus, in the training process, the exchange of information between tasks and the competition between domains can improve the performance of each task.

In [35], the MTL of deep learning networks was divided into two categories: hard and soft parameter sharing. Hard parameter sharing primarily uses shared hidden layers in a deep learning network with multiple task outputs, as shown in Figure 5a. This approach can force the network to find a more generalized model that can satisfy all tasks during training and is easy to overfit. Soft parameter sharing means that different tasks have independent models and parameters; however, each model mutually restricts the distance between its parameters during training, as shown in Figure 5b. This method can encourage each model to improve each other's similarities, but they retain the ability to complete the specified task. There are many ways to calculate the distance, such as the Euclidean distance or trace norm.

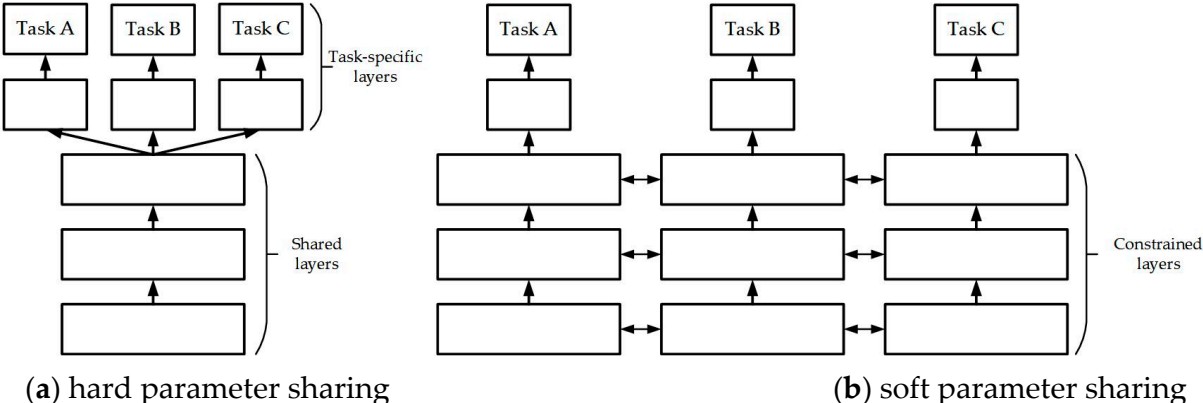

(**a**) hard parameter sharing          (**b**) soft parameter sharing

**Figure 5.** Parameter-sharing diagram.

An inductive bias obtained via MTL seems intuitively plausible. However, the following features highlight the advantages of this approach. First, MTL can indirectly increase the amount of training information. The second feature is that this method allows the deep learning network to focus on more-valuable features during training. The third feature is that MTL causes the network to exhibit a representative bias when learning. In addition, the MTL can import inductive bias during in-depth learning network training to make the network model less easy to overfit. Finally, it can also add an eavesdropping training mechanism to make it easier for the network to learn different tasks.

### 3.1. Multidomain Joint Learning

The MDJL approach proposed in this paper addresses the following problem: when a multidomain dataset is used to train a deep learning network, the final model performs worse on individual domains than on separate training on single domains. As defined in [34] for MTL, if we treat pedestrian detection in different domains as a different task, we can treat the aforementioned problem as an MTL problem.

The utilized MTL method has the hard parameter-sharing architecture described in [35]. It includes a shared feature extraction network, a task-specific hidden layer, and the output of each task. However, given that all the outputs have the same class probability and bounding box, we combine and share the outputs of various tasks. Figure 6 shows a simple architectural diagram. The colored blocks in the figure represent task-specific hidden layers. The objective is that the network will independently construct each task block during the training process. Therefore, we used DGD to facilitate the self-allocation of hidden layer resources in the network, according to the response of neurons to different tasks in the hidden layer.

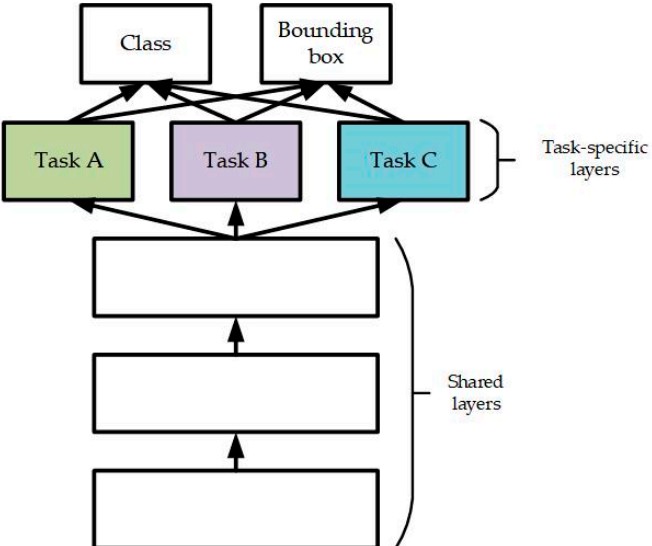

**Figure 6.** MDJL hard parameter-sharing architecture diagram.

MDJL divides the classification network into the same number of subnetworks according to the number of domains in the multidomain dataset. The division standard was determined on the basis of the response strength of each neuron in the classification network of different domains. Thus, each subnet can be specialized in a specific domain to more efficiently learn the information of various domains contained in the multidomain dataset and improve the network's overall performance in each domain.

To calculate the degree of response of a specific neuron to a specific domain, we used the NIS in the DGD, which is defined in Equation (1) as follows:

$$s_i = L\big(g(x)_{\setminus i}\big) - L(g(x)),\tag{1}$$

where $g(x)_{\setminus i}$ is the output of the entire network when the $i$th neuron is ignored, $g(x)$ is the original output, and $L(x)$ is a loss function. Because our network has two outputs, a bounding box and category probability distribution, we use the multitask loss to evaluate the network's performance.

A DGD has two modes: deterministic and stochastic. Equation (2), which expresses the deterministic mode, is used to drop out all neurons with a nonpositive impact score (NIS) of less than zero.

$$\begin{cases} keep, \ if \ s_i > 0 \\ dropout, \ if \ s_i \le 0 \end{cases}\tag{2}$$

The stochastic mode uses the sigmoid function to determine the dropout probability of a neuron, which is defined in Equation (3) as follows:

$$p(s_i) = \frac{1}{1 + e^{-s_i/T}},\tag{3}$$

where $T$ is an adjustable parameter, and its value determines the degree of the NIS influence probability. If $T \to 0$, then it becomes deterministic. If $T \to \infty$, it returns to the traditional dropout. In this paper, MDJL adopts the deterministic mode in the preliminary training stage, whereas the stochastic mode is used for fine-tuning.

Given that MDJL has two types of outputs (class and bounding box), the bounding box's output is meaningful only when the object is classified as nonbackground to facilitate end-to-end training in the network and obtain the best parameters $W^*$. Therefore, multitask loss was used as the evaluation basis for training the network. The sample set used in training was $S = \{(X_i, Y_i)\}_{i=1}^{K}$, where $X_i$ is the sample image, $Y_i = (y_i, b_i)$ is the category $y_i \in \{1, 2, 3, \dots, K\}$ to which the sample belongs, and the bounding box position of the

actual object is $b_i = \left( b_i^x, b_i^y, b_i^w, b_i^h \right)$. Using this information, we can write the multitask loss function as follows, in Equation (4):

$$(\boldsymbol{W}) = \sum_{m=1}^{M} \sum_{i \in S^m} \alpha_m l^m (X_i, Y_i | \boldsymbol{W}), \qquad (4)$$

where $M$ is the number of branches, $\alpha_m$ is the weight of $l^m$, and $S^m$ in $S = \left\{ S^1, S^2, \ldots, S^M \right\}$ is the collection of objects in charge of this branch. Thus, $l^m$ calculates the loss from two aspects: the correctness of classification and the accuracy of the bounding box. It takes the form of Equation (5):

$$l(X, Y | \boldsymbol{W}) = L_{cls}(p(X), y) + \lambda [y \geq 1] L_{loc} \left( b, \hat{b} \right) \qquad (5)$$

$p(X) = \{ p_0(X), p_1(X), \ldots, p_K(X) \}$ is the probability distribution of each category, and $L_{cls}$ is the classified cross-entropy loss, which is defined in Equation (6) as follows:

$$L_{cls}(p(X), y) = -\log p_y(X) \qquad (6)$$

$\hat{b} = \{ b_x, b_y, b_w, b_h \}$ are the calculated bounding boxes. MS-CNN uses Equation (7), defined in [26], to calculate the bounding box loss.

$$L_{loc} = \frac{1}{4} \sum_{j \in \{x,y,w,h\}} smooth \left( b_j, \hat{b}_j \right) \qquad (7)$$

From Equation (5), $L_{loc}$ is valid only for nonbackground categories, and $\lambda$ is the weight parameter. After the error has been obtained, the best parameter $\boldsymbol{W}^* = argmin_{\boldsymbol{W}} \, \mathcal{L}(\boldsymbol{W})$ can be obtained using the stochastic gradient descent method.

To equitably calculate the NIS of each domain, it is necessary to calculate the NIS of all validation samples in the domain, followed by the calculation of the average to obtain the final score. Figure 7 shows a schematic of the MDJL network. To replace the initially used dropout with DGD in the MS-CNN and reduce the error caused by the difference in the implementation, we adopted the DGD three-stage training method. The detailed training process is explained in Section 3.3.

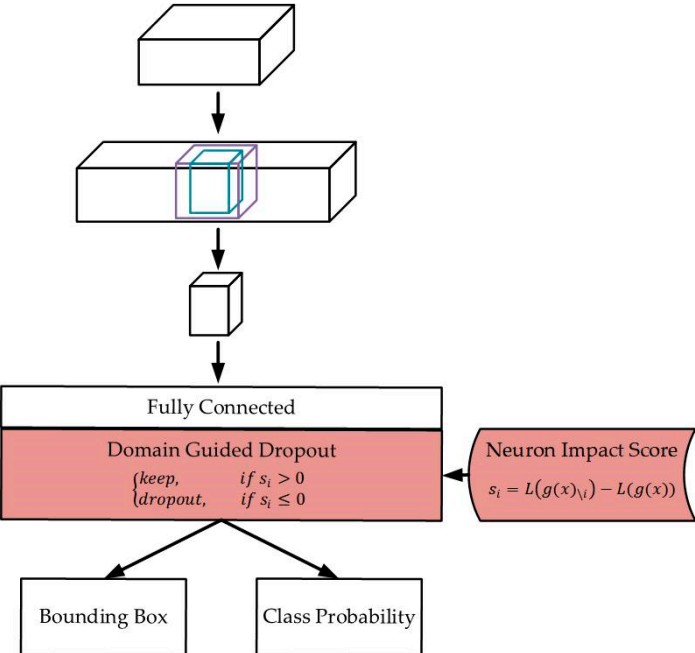

**Figure 7.** MDJL schematic.

### 3.2. Domain-Oriented Fine-Tuning

To further improve the network's performance, the consensus in the research community is to fine-tune a preliminarily trained network. In this section, domain-oriented fine-tuning is presented as a method that uses domain information to effectively fine-tune the MDJL.

We use Equation (1) to calculate the NIS of each neuron for each domain and use the rules of Equation (2) to allocate the domain for which the neuron is responsible. Because the same neuron may be responsible for more than one domain, we can represent the distribution of the neurons that are responsible for different domains by using a Venn diagram, as shown in Figure 8. This figure shows that each domain has unique neurons (red, yellow, and blue blocks). These neurons are known as domain-exclusive neurons (DENs). DENs are responsible for only a single domain; therefore, it is feasible that fine-tuning these neurons can further improve the network's performance.

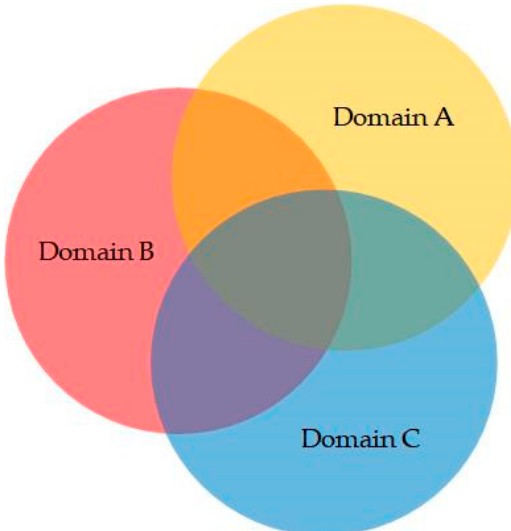

**Figure 8.** Schematic of neuron distribution in MDJL according to NIS classification.

By using the characteristics of stochastic DGD, the drop rate can be significantly increased during the training of neurons that do not belong to DENs by manipulating the NIS of each neuron for different domains. Thus, the fine-tuned DENs in the network can be further improved. To obtain the DENs $\varepsilon$ that belongs to each domain, we used the previously calculated NIS score $\boldsymbol{\mathcal{S}}_j = \{s_{1j}, s_{2j}, s_{3j}, \ldots, s_{ij}, \ldots, s_{nk}\}$ for all $n$ neurons for all $k$ domains $\boldsymbol{\mathcal{D}} = \{\mathcal{D}_1, \mathcal{D}_2, \mathcal{D}_3, \ldots, \mathcal{D}_j, \ldots, \mathcal{D}_k\}$ and Equation (2) to determine the neuron set $\mathcal{N}_j$ responsible for domain $\mathcal{D}_j$. Then, if $I = \{1, 2, 3, \ldots, k\}$, we obtain the DENs $\varepsilon_j = \mathcal{N}_j \backslash (\cup_{i \in I \ \& \ i \neq j} \mathcal{N}_i)$ that belong to domain $\mathcal{D}_j$. Finally, we used Equation (8) to obtain the adjusted NIS of each neuron $n_i$, called the exclusive neuron impact score (eNIS). $R$, known as the reserve constant, is a parameter less than 0 that can be adjusted to modify the reserve rate of neurons that do not belong to the DENs so that it is smaller than the reserve rate of neuron DENs. The reserve rate used in fine-tuning is the result calculated using Equation (3) and using eNIS, as shown in Figure 9.

$$\hat{s}_{ij} = \begin{cases} s_{ij}, \ if \ n_i \in \varepsilon_j \\ R, \ otherwise \end{cases} \tag{8}$$

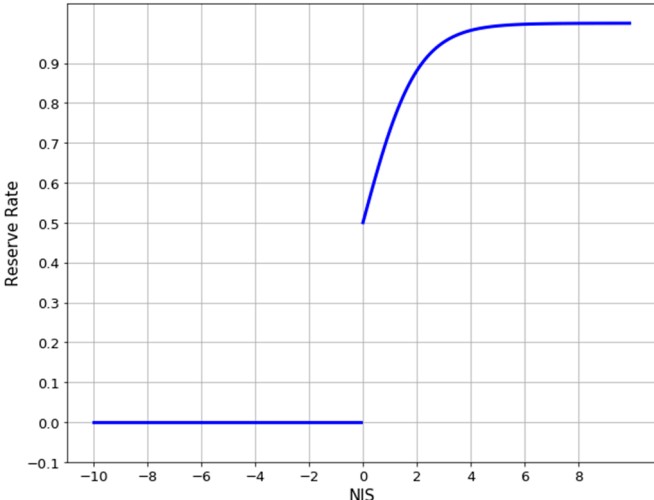

**Figure 9.** Adjusted reserve rate.

*3.3. Multidomain Joint Learning Training*

Backpropagation was used as the training algorithm. Training of the MDJL was divided into three stages. The first stage synthesized the initial training of all the datasets. We used VGG16 as our initial model to obtain a basic network that had learned all the domains. The second stage analyzed the NIS of the basic network obtained in the first stage, replaced the dropout in the network with DGD, and performed subsequent training actions. In this stage, the DGD dropped the specified neuron according to the NIS of the domain to which the input sample belonged during the training process, thereby strengthening the network's concept of the domain to improve its performance. In the final stage, we propose domain-oriented fine-tuning to optimize the network model further.

The network architecture used in this paper was MS-CNN; therefore, the first stage of initialization training was conducted according to the MS-CNN training method. This method also has two training stages. The main difference between the two stages is the negative sampling mode. The first stage adopts the random sampling mode, and the second stage adopts the bootstrapping sampling mode. Bootstrapping is a type of hard-negative mining-sampling method. It can be used to determine whether a sample can be used for training according to the objectness score of the sample. Its goal is to facilitate easier misjudging of the collected sample, thereby reducing the network chance of misjudgment.

During training, the training sample set of the branch detection layer $m$ is defined as $S^m = \{S^m_+, S^m_-\}$, where $S^m_+$ represents positive samples and $S^m_-$ represents negative samples. Assume that a single sample is defined as $S = (X_i, Y_i)$, where $X_i$ is the sample image and $Y_i = (y_i, b_i)$ is the category to which the sample belongs and the location of the bounding box. Here, $o^*$, as defined in Equation (9), determines whether the sample is positive or negative.

$$o^* = \max_{i \in S_{gt}} IoU(b, b_i) \tag{9}$$

$S_{gt}$ is the ground truth, and $IoU$ is the intersection between the two bounding boxes. Thus, when $o^*$ is greater than or equal to 0.5, the sample is added to the positive sample set; when $o^*$ is less than 0.2, the sample is added to the negative sample set, and all other samples are discarded. Typically, the distribution of objects and nonobjects in an image is asymmetric. Therefore, to achieve a balance between the two, we set $|S^m_-| = \gamma |S^m_+|$. Random sampling and bootstrapping sampling are then used to obtain the final negative sample set.

To ensure that each detection layer detects objects only within a fixed scale range, the training set of each layer is composed of a subset of $S$, and the samples of this subset correspond to the scale range of the layer detection. However, this method may cause a specific layer to have no available positive samples and indirectly cause learning instability.

Therefore, it is necessary to modify the cross-entropy, as shown in Equation (10), to solve this problem.

$$L_{cls} = \frac{1}{1+\gamma} \frac{1}{|S_+|} \sum_{i \in S_+} -log p_{y_i}(X_i) + \frac{1}{1+\gamma} \frac{1}{|S_-|} \sum_{i \in S_-} -log p_0(X_i) \tag{10}$$

### 3.4. Multiobject Tracking System for Flying Drones

The operation process of the proposed online multiobject tracking system is shown in Figure 10. It can be divided into two parts: the object detector and the tracker. The MDJL algorithm proposed in this paper is used for the former, whereas the MDP tracker algorithm is used for the latter. The MDP tracker is an online tracking algorithm that can instantaneously calculate and output results by using new data input. These algorithms are more suitable for applications that require real-time feedback, applications such as flying drones.

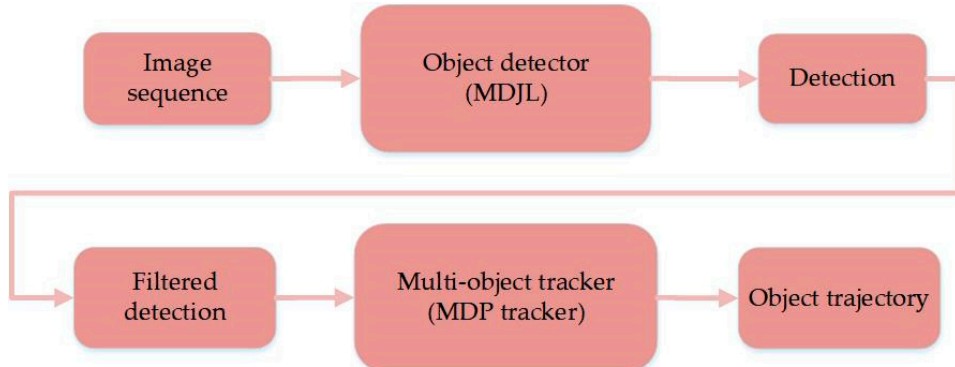

**Figure 10.** Flow chart of multiobject tracking system.

Our online tracking system utilized MDJL as the pedestrian detector, and the MDP was used as the tracker. By combining the pedestrian detection methods proposed in this paper, it is expected that the multiobject tracking system will be more effective in multidomain situations. In an MDP tracker, each object builds an MDP model. This model is composed of four elements (), where $S$ represents the state of the object; $A$ and $T(\cdot)$ represent the action and transfer equations executed when the object is transferred, respectively; and $R(\cdot)$ represents the reward function after the $A$ action has been performed.

In the MDP model, S has four states: active, tracked, lost, and inactive. Figure 11 is a flowchart of the status of the MDP tracker. First, when the object detector detects an object, it is initialized and enters the active state. In this state, it determines whether subsequent tracking should be performed and changes the status to either tracked or inactive. When the status is tracked, the target is continuously tracked and maintained in the tracked status. However, when tracking is not possible, such as when the target is obscured or exits the field of view of the camera, the MDP changes the status to lost. If the target appears again and is successfully associated, the state is changed to tracked; otherwise, it remains lost. When the status is lost for more than a period, the MDP changes to an inactive status. Moreover, the MDP aborts all tracking actions and does not change the status.

During tracking, each target has an appearance model consisting of several appearance templates that were created during the tracking process. These templates record the appearance of the target during tracking. To maintain the validity of the appearance model, they must be updated. Therefore, these templates adopt a lazy-update rule. It is updated for the current appearance only when the tracked target status changes from lost to tracked.

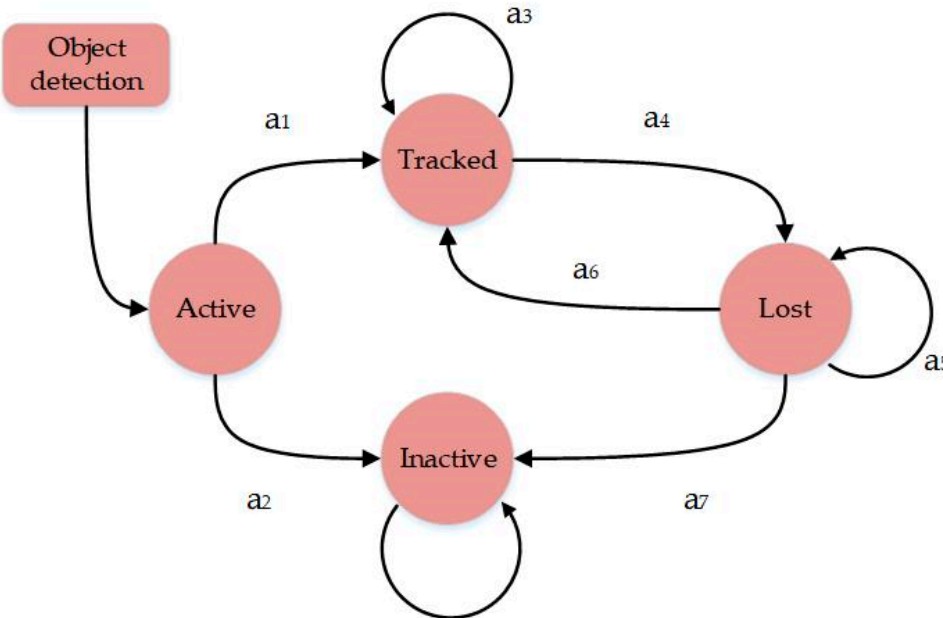

**Figure 11.** Flowchart of MDP tracker status.

The decision on the lost state can be considered as a data association problem. Therefore, the MDP tracker continues to be associated with the new bounding box in the new frame in this state. If there is an association with the bounding box, it changes the state to track; otherwise, it continues to maintain the lost state. When the lost state is maintained for more than a specific frame number, the target is considered to have left the camera's field of view, and the state is changed to inactive, to terminate tracking.

During the training process, the MDP adopts reinforcement learning. First, given multiple video sequences as training data, it can track all objects in these videos, and the training goal is to facilitate the successful tracking of all objects. The MDP tracks the target according to this strategy for each state, uses the reward equation to evaluate the performance of the tracker, and maximizes the result of the final tracker's reward equation by adjusting the parameters of the reward equation for the lost state. The training algorithm processes all the training videos and tracks the training targets in a loop, and the training does not end until all the targets have been successfully tracked. During training, data association is updated only when an error occurs during this process. There are two possible errors: the first case is associated with a nontarget bounding box, and the second case occurs when a target bounding box appears but is not associated. When these errors occur, an incorrect association is added to the negative training sample. Conversely, a correct association was added to the positive training sample for training.

## 4. Experimental Results

The experiment was conducted in two parts. In the first part, we evaluated the performance of our proposed MDJL pedestrian detection method. In the second part, we adjusted the pedestrian tracking parameters on the basis of our MDJL for comparison with other advanced tracking algorithms.

The computer hardware specifications used in the experiment included an Intel Core i7-930 CPU with 24 GB RAM and an Nvidia GTX 1080Ti GPU with 11 GB RAM. The operating system was Ubuntu 14.04 LTS, and CUDA 8.0, CUDNN 5.1, and MATLAB R2014b were the primary software programs.

In the experiment, we used datasets primarily from three sources. The Caltech pedestrian detection dataset [36] (Caltech dataset) is a driving video. The format of the video is $640 \times 480$ pixels at 30 Hz. This dataset includes images with approximately 250,000 frames, 350,000 bounding boxes, and 2300 unique pedestrians. Among them, set00–set05 was defined as the training dataset, and set06–set10 was defined as the testing dataset. To

ensure data integrity, set05 in the training dataset was excluded from the validation dataset that was used during training. The shooting angle used in this dataset was of a forward orientation, as shown in Figure 12a.

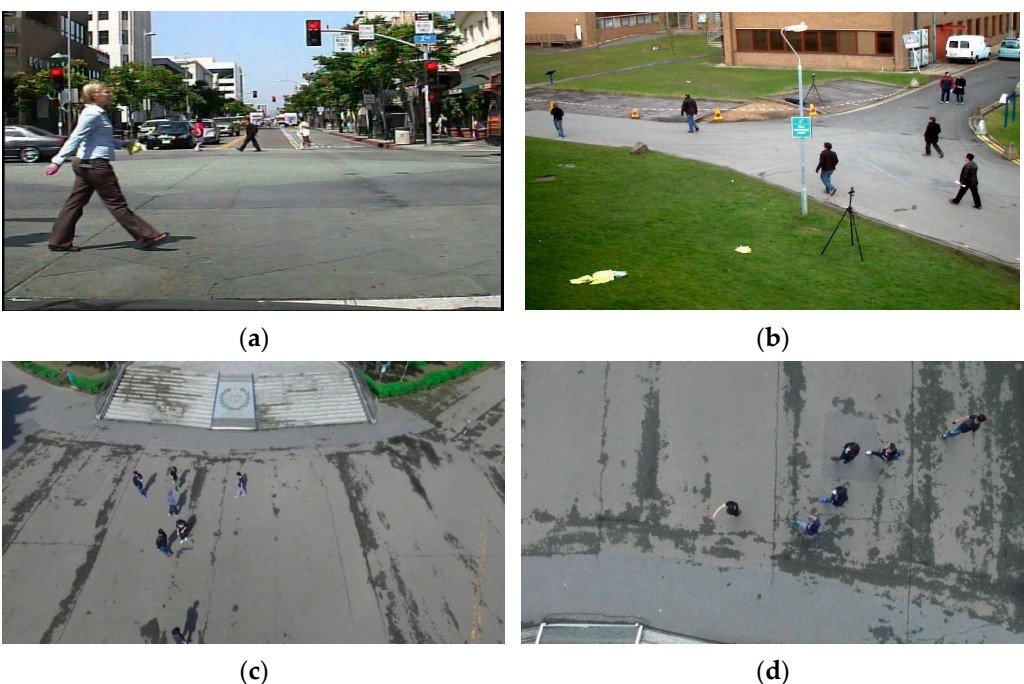

**Figure 12.** Different shooting angles of each dataset: (**a**) Caltech dataset—forward orientation angle, (**b**) PETS09 dataset—top view angle, and (**c**) DroneFJU—45-degree downward and (**d**) 90-degree downward orientations.

The primary purpose of the PETS09 dataset [37] was to track an individual within a crowd and to detect flow and crowd events. Eight cameras and eight angles were used to capture a designated area at the same time on the campus. The video format used was 768 × 576 pixels at 7 Hz. In this paper, to prevent the PETS09 and Caltech datasets from using videos with repeated shooting angles, only the downward-oriented (top view angle) videos in PETS09 were used, as shown in Figure 12b. We divided these seven videos into three sets: training (S1L2-1, S2L1, S2L3, and S3MF1); validation (S1L1-1 and S1L1-2); and testing (S2L2).

The DroneFJU dataset was obtained on the Fu Jen Catholic University campus using a quadrotor. The equipment included a DJI Matrice 100 + Zenmuse-integrated PTZ camera. The shooting format was 1080 × 1920 pixels at 60 Hz, and the shooting angle was divided into a 45-degree downward-facing orientation (Figure 12c) and a 90-degree downward orientation (Figure 12d). The latter shooting angle was the challenging aspect of this dataset. In addition, because a wide-angle lens was used for video acquisition, the pedestrian's face and overall outline changed depending on their position (as shown in Figure 12d, red box) and various body postures.

In Section 4.1, the performance of the proposed MDJL pedestrian detection method is analyzed by using an ablation study; moreover, the network characteristics of the MDJL are analyzed in Section 4.2. In Section 4.3, domain-oriented fine-tuning is implemented, and its ability to improve network performance is evaluated. In Section 4.4, we discuss and optimize the parameters of domain-oriented fine-tuning and compare it with the current state-of-the-art pedestrian detection method.

Finally, we used the PET09 and the TUD datasets with pedestrians as tracking objects and compared them with the original MDP and several more advanced tracking methods. In Section 4.5, we introduce the evaluation methods and present the adjustments we performed to facilitate pedestrian detection, in addition to the influence of the pedestrian

tracking parameters on the final results. We present the optimization results of our method using an experimental video and a comparative analysis with other tracking methods.

### 4.1. Ablation Study

MDJL is a network training method designed for a multidomain dataset. Our objective is to use this method to address the reduction in network accuracy that occurs when using a multidomain dataset for training. In this experiment, we discuss the aforementioned issues and evaluate the effectiveness of our method.

In the following experiments, we used the Caltech dataset (referred to as c), PETS09 (referred to as p), and DroneFJU (referred to as d) for different sizes and different shooting methods, to combine them into several multidomain datasets and use these datasets to train the baseline network (MS-CNN) and our proposed method (MDJL). The method of evaluating the network uses an appropriate setting of the Caltech pedestrian detection benchmark to assess the network's average miss rate and the receiver-operating characteristic (ROC) curve in the test set of the aforementioned three datasets.

Table 1 summarizes the number of training, validation, and testing samples for each dataset. The number of training samples of PETS09 is approximately one-third of that of the Caltech dataset, whereas the number of training samples of DroneFJU is approximately one-half of that of the Caltech dataset. This difference may affect the final network performance. Therefore, in this experiment, it was necessary to increase the number of training samples of the smaller datasets to the same or more than that of the Caltech dataset. We used two data augmentation methods to create two additional PETS09 training images and a DroneFJU training image. The first method flips the original image left and right, and the second method adds Gaussian noise to the original image. The Gaussian noise parameters had an average of 0 and a standard deviation of 0.01. Using these two data augmentation methods, we increased the number of training samples of PETS09 to 36,504, whereas that of DroneFJU increased to 35,632 via the addition of Gaussian noise. Finally, to balance the number of samples in each dataset, we used the number of training samples of the Caltech dataset as the standard and randomly extracted the same number of samples from the expanded dataset to obtain the final dataset.

**Table 1.** Number of dataset samples.

| Dataset Name | Sample Count | | |
| --- | --- | --- | --- |
| | Training | Validation | Testing |
| Caltech dataset | 34,927 | 3346 | 33,043 |
| PETS09 | 12,168 | 8813 | 14,474 |
| DroneFJU | 17,816 | 4286 | 6460 |

Eight types of multidomain datasets were used, and the abbreviations used for the new dataset combine the names of each. Therefore, for example, for Caltech + PETS09, the multidomain dataset is called cp, and for Caltech + PETS09 + DroneFJU, the dataset is called cpd. If the multidomain dataset used data augmentation to balance the number of samples, the symbol * was used next to the abbreviation. For example, Caltech + PETS09 with data augmentation is denoted as cp*, and Caltech + PETS09 with data augmentation + DroneFJU with data augmentation is denoted as cp*d*.

First, let us examine the evaluation results for different datasets when we use a single dataset, the so-called single-domain dataset, to train the model. We used the Caltech dataset, PETS09, and DroneFJU to train the MS-CNN model and to evaluate the three trained models, c, p, and d, on the test set of each dataset. The results are shown in Figure 13.

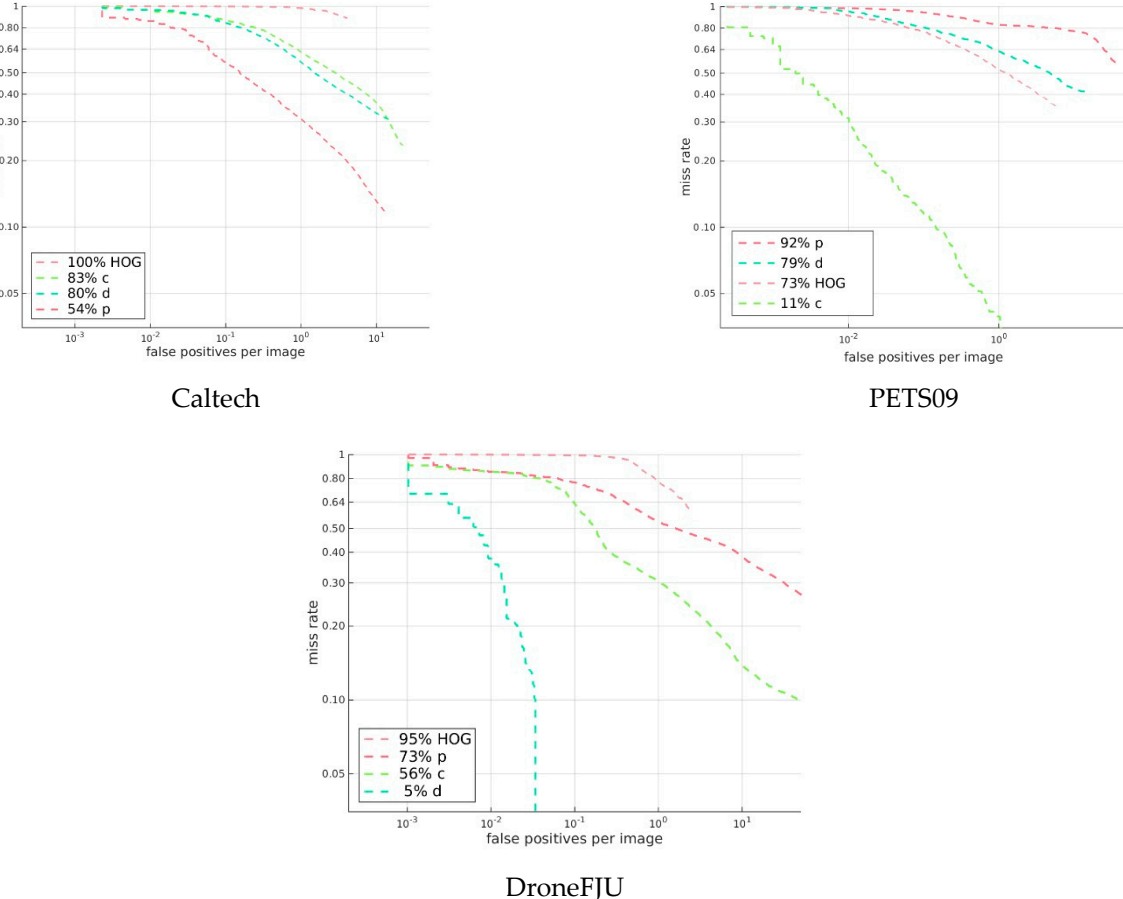

**Figure 13.** Evaluation results of single-domain dataset model in different datasets.

Figure 13 shows that all the single-domain dataset models yield better evaluation results only for the dataset of the same domain and perform poorly on other domains. These results show that a model trained using a single dataset can be applied only to the same domain and cannot be generalized to other domains.

Table 2 lists the test results of the models trained using different multidomain datasets with appropriate settings of the test set of the Caltech dataset. We also include the evaluation results for the same setting when MS-CNN uses only the Caltech dataset as the baseline for comparison. The corresponding data are not shown for MDJL because they form a network designed for a multidomain dataset. Therefore, we did not consider the performance when using a single dataset for training. To perform a fair comparison, we selected only the multidomain dataset containing the Caltech dataset.

**Table 2.** Average miss rates of different multidomain dataset models for the Caltech test set.

| Training Dataset | Average Miss Rate | |
| --- | --- | --- |
| | **MS-CNN** | **MDJL** |
| cp* | 0.12103 | **0.114468** |
| cp*d* | 0.115921 | **0.113062** |
| cd* | 0.11407 | **0.113394** |
| cpd | 0.113918 | **0.106365** |
| cp | 0.112784 | **0.112716** |
| cd | 0.112092 | **0.108718** |
| c | 0.110483 | |

The MS-CNN section is summarized in Table 2. When training was performed with the cp, cd, and cpd datasets without data augmentation, it was determined that simply adding a domain (such as cp and cd) to the dataset increased the average miss rate by approximately 0.2%, whereas adding cpd to the three domains increased it by approximately 0.5%. If we examine cp*, cd*, and cp*d* that use data augmentation, it is determined that the average miss rate further increased by approximately 1%. Thus, the overall average miss rate increased by 0.5%, with a standard deviation of 0.006. This indicates that the greater the number of domains involved in training the network with a multidomain dataset, the more significant the impact on the network performance. Moreover, using data augmentation to balance the number of samples increased the influence of the initially small dataset on the network training and also affected the accuracy of the network.

The experimental results of MDJL show that regardless of the number of domains contained in the multidomain dataset, the average miss rate increased by only approximately 0.2%. Furthermore, even if data augmentation was used, the average miss rate increased by only approximately 0.3%. Thus, the overall average increase in the average miss rate was 0.1%, and the standard deviation was 0.002. This indicates that MDJL is more stable when using multidomain dataset training and is not highly susceptible to the number of domains or the number of data. In addition, regardless of the multidomain dataset used for training, the average miss rate of MDJL was lower than that of MS-CNN. This implies that MDJL can improve the overall accuracy when training with a multidomain dataset.

In the following table, we compare the performance of the models trained using each multidomain dataset on the test set. Therefore, we indicate the dataset name used in the training after the method name to facilitate comparison. For example, the MDJL model trained with the cpd dataset is called MDJL-cpd.

We used the MS-CNN and MDJL models, which were trained on the eight multidomain datasets, to evaluate the test sets of the Caltech, PETS09, and DroneFJU datasets, and we recorded the average miss rates in Table 3. Two aspects were analyzed in this table. First, the performance of the MS-CNN and MDJL models that were trained using the same multidomain dataset for each evaluation dataset is identified by using a thick box. We investigated the network trained using the three datasets of cpd, cd, and pd, and it was determined that the average miss rate evaluated for PEST09 and DroneFJU using MDJL was higher than that of MS-CNN. None of these datasets used data augmentation to balance the sample size difference between the domains. Therefore, we speculate that even if our method can effectively divide the network into several small networks for fine-tuning on the basis of the response to each domain, as shown in Figure 14, shared and generalized neurons will still exist across the domains. These neurons are more susceptible to larger datasets (Caltech) during training and are dominated by these datasets, thus affecting the network performance on smaller datasets (PETS09 and DroneFJU). However, suppose we use the same domain combination of cp*d*, cd*, and p*d* for data augmentation. In this case, this problem is not encountered, which establishes that our assertion is reasonable. To optimize and balance the network's performance on each dataset, the number of samples of each domain in the multidomain dataset must also be balanced. Therefore, when discussing other phenomena, we will use the data augmentation model.

Second, the three blocks in Table 3 that are identified by using a gray color represent the network trained by each multidomain dataset. The performance on the test set of the dataset is interesting in each case. The results indicate that the model was trained using two datasets, cp* and cd*. It is foreseeable that the evaluation results of these models on the datasets, which have never been observed, are not satisfactory. However, the data show that MDJL-cp* exhibits an improvement of 2.7% in the average miss rate compared with MS-CNN-cp* and that MDJL-cd* is 1.7% better than MS-CNN-cd*. This improvement is not insignificant, so we believe that the model trained by MDJL has better generalizability.

**Table 3.** Average miss rate for each model for each dataset.

| Model | Average Miss Rate | | |
|---|---|---|---|
| | Caltech | PETS09 | DroneFJU |
| MS-CNN-cpd | 0.113918 | **0.537132** | **0.080445** |
| MDJL-cpd | **0.106365** | 0.546494 | 0.087827 |
| MS-CNN-cp*d* | 0.115921 | 0.498298 | 0.054163 |
| MDJL-cp*d* | **0.113062** | **0.487953** | **0.051723** |
| MS-CNN-cp | 0.112784 | 0.566055 | 0.578502 |
| MDJL-cp | **0.112716** | **0.556025** | **0.516990** |
| MS-CNN-cp* | 0.121030 | 0.523379 | 0.444892 |
| MDJL-cp* | **0.114468** | **0.517952** | **0.417621** |
| MS-CNN-cd | 0.112092 | **0.764734** | **0.079890** |
| MDJL-cd | **0.108718** | 0.794011 | 0.100103 |
| MS-CNN-cd* | 0.114070 | 0.799884 | 0.061482 |
| MDJL-cd* | **0.113394** | **0.782861** | 0.055286 |
| MS-CNN-pd | **0.782319** | 0.517785 | **0.020215** |
| MDJL-pd | 0.801515 | 0.538508 | 0.022154 |
| MS-CNN-p*d* | **0.805207** | 0.529371 | 0.024825 |
| MDJL-p*d* | 0.810766 | **0.519591** | **0.018935** |

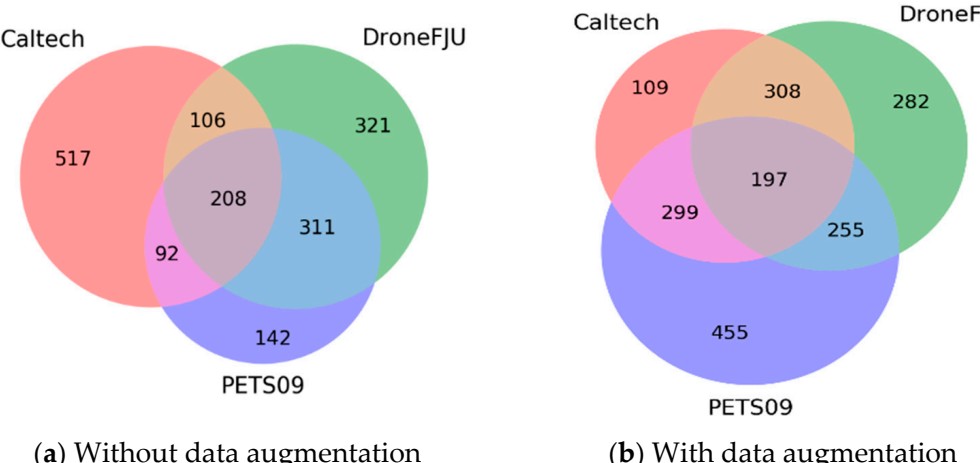

(**a**) Without data augmentation      (**b**) With data augmentation

**Figure 14.** Degree of subnetwork overlaps of each domain.

However, if we examine the MS-CNN-p*d*and MDJL-p*d*models trained using the p*d* dataset, this phenomenon is not observed, and the evaluation performance of the two networks is similar. This result is not surprising given that the ability to generalize originates from similar situations that are learned or seen, from which knowledge is extended. The two datasets, PETS09 and DroneFJU, differ from the Caltech dataset used for the evaluation of the shooting angle and the shooting methods, which results in different image characteristics in the dataset. Therefore, neither MS-CNN nor MDJL can be used in PETS09, and DroneFJU learned how to correctly evaluate the Caltech dataset to obtain the presented results.

According to the results of this experiment, it is evident that the proposed method can effectively improve the impact of training networks using a multidomain dataset. Furthermore, using data augmentation to balance the number of samples in each domain in a multidomain dataset can increase the effectiveness of MDJL. Finally, MDJL appears to have better generalizability than MS-CNN does.

### 4.2. Analysis of Multidomain Joint Learning Method

In this experiment, the NIS calculated using MDJL is represented as a chart to analyze the relationship between the subnetworks after segmentation. First, we investigated whether there was a relationship between the response degree (NIS) of each neuron to different domains in the network. Three sets of comparisons were conducted, each of which selected two domains, A and B. We then sorted the NIS of domain A from small to large, arranged domain B in the same order, and represented the results as a line chart. As shown in Figure 15, domain A is the blue curve, and domain B is the red curve. The three results in the figure show that the NIS among the various domains does not exhibit a correlation, which means that the neurons in the network that strongly respond to each domain are not the same.

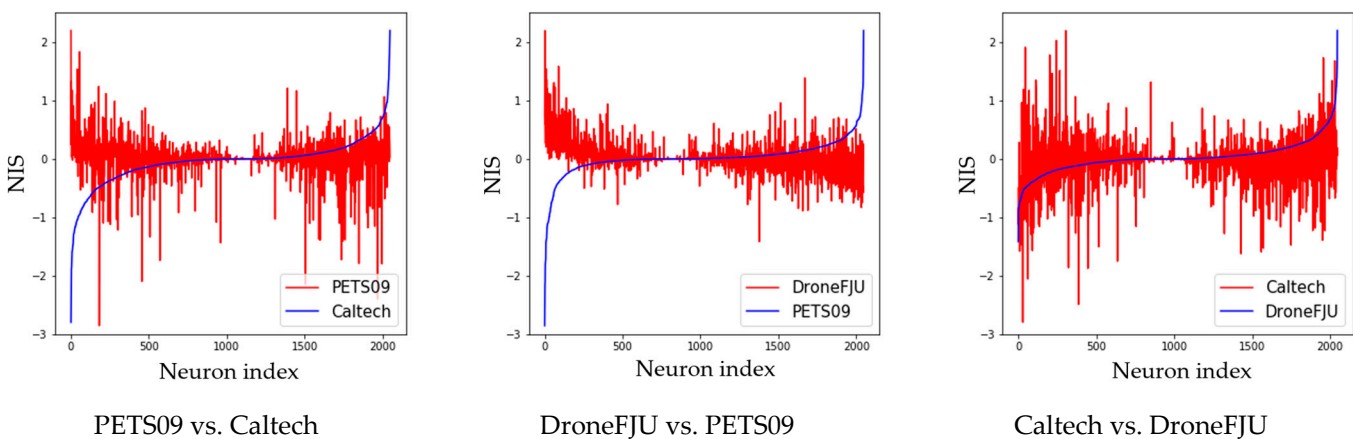

PETS09 vs. Caltech      DroneFJU vs. PETS09      Caltech vs. DroneFJU

**Figure 15.** NIS correlation between different domains.

Next, we analyze the degree of overlap in the subnetworks divided according to the deterministic rule (NIS > 0), according to the NIS of each domain, and the results are represented as a Venn diagram. Figure 14 shows the distribution of neurons according to the models trained using the datasets with and without data augmentation.

First, in Figure 14a, a dataset without data augmentation is used to train the model, and the distribution of neurons is allocated to more or fewer neurons according to the training sample size of each domain.

Thus, a large domain (e.g., Caltech) is assigned to more neurons, and small domains (e.g., PETS09) are assigned to fewer neurons. This phenomenon is consistent with the observations summarized in Table 3. Therefore, large domains dominate the network. In Figure 14b, the neurons are distributed according to the complexity of the domain because of data augmentation. PETS09 was used because the pedestrians in the video were sometimes dense and sometimes loose. The shooting angle is a downward projection; it can easily cause pedestrian deformation, and the situation is complex. Therefore, when the network allocates neurons, it provides more neurons to the PETS09 domain. In contrast, although the scene at Caltech is relatively open because the camera's shooting angle is forward facing and the situation when pedestrians appear is relatively simple, the network allocates fewer neurons to this domain.

In addition, from Figure 15, it is evident that although there is no correlation between the NIS of each domain, there are still neurons in Figure 14 included in the multidomain subnets. We believe that these neurons are responsible for common information processing across domains. Moreover, the nonoverlapping parts are responsible for the unique information processing of the domain, thereby achieving the concept of divide and conquer. From the aforementioned two results, it is evident that MDJL can effectively divide the network into several subnetworks according to the degree of the neuron response to different domains. It includes dedicated neurons for each domain and neurons shared among the domains.

### 4.3. Domain-Oriented Fine-Tuning Experiments and Analysis

In this experiment, we used the previously trained model MDJL-cp*d* to perform further fine-tuning and obtain the model's evaluation results for each dataset, as summarized in Table 4. The gray background is the performance evaluated by the model trained using the single-domain dataset in the corresponding dataset. Therefore, for the evaluation, only Caltech assessed MS-CNN-c, PETS09 only assessed MS-CNN-p, and MS-CNN-d only used DroneFJU. We named this model MDJL-ft-cp*d*.

**Table 4.** Average miss rate evaluated in each dataset after fine-tuning.

| Model | Average Miss Rate | | |
|---|---|---|---|
| | Caltech | PETS09 | DroneFJU |
| MS-CNN-c/-p/-d | 0.110483 | 0.544825 | 0.053416 |
| MDJL-cp*d* | 0.113062 | 0.487953 | 0.051723 |
| MDJL-ft-cp*d* | **0.107807** | **0.482616** | **0.046973** |

According to Table 4, it is evident that the average miss rate of the model after fine-tuning for the evaluation using the Caltech dataset is 10.7807%, which is better than that of MS-CNN's 11.0483%. This is a slight improvement on PETS09 and DroneFJU. These results show that domain-oriented fine-tuning can effectively strengthen the domain-exclusive neurons and improve the network model. Therefore, we discuss in the next section how to adjust the parameters of domain-oriented fine-tuning.

There are three adjustable parameters for domain-oriented fine-tuning. The first is the T parameter in Equation (3), which we call the sigmoid scaler. The second is the R parameter in Equation (8), which we refer to as the reserve constant. The third is the number of iterations during fine-tuning training. We separately explore how the adjustment of these three parameters affects the performance of the network. The parameters that did not change in the following experiments were preset values: T = 1.0, R = 0, and iteration number = 20,000.

First, we discuss the T parameter in Equation (3), which is responsible primarily for adjusting the influence of the eNIS on the reserve rate of domain-exclusive neurons during fine-tuning. When $T \to 0$, stochastic DGD is converted to deterministic DGD because it can convert the result of Equation (3) to only 1 (eNIS $\geq$ 0) or 0 (eNIS < 0). In contrast, if $T \to \infty$, regardless of the value of eNIS, the result of Equation (3) is always 0.5, causing stochastic DGD to degenerate back to regular dropout. For convenience, we used the $1/T$ value as the adjustment target in the following experiments.

We adjusted $1/T$ from 0.1 to 1.0 and trained 10 models using 0.1 as the interval in the experiment. Each trained model was evaluated using the Caltech test set, PETS09, and the DroneFJU datasets, and the average of the results was calculated. The experimental results are shown in Figure 16a.

Figure 16a indicates that there is no consistent trend in the performance of different $1/T$ parameters for each dataset. However, if we observe the overall average performance in Figure 16a, we find that the relationship between the parameter T and the miss rate is slightly low, in the range of 0.2–0.5. Thus, the lowest point was at $1/T$ = 0.5. According to Section 4.2, the maximum NIS value is approximately 2. Therefore, by applying Equation (3), the highest reserve rate of domain-exclusive neurons in fine-tuning is 73.11%.

The second parameter to be discussed is the R parameter in Equation (8), namely the reserve constant. This parameter is responsible for adjusting the reserve rate of neurons that do not belong to domain-exclusive neurons in domain-oriented fine-tuning and is a number less than 0. We adjusted the parameter R from 0 to −10, with −1 as the interval, and changed it 11 times. Each trained model was evaluated by using the Caltech test set, PETS09, and the DroneFJU datasets, and the average was calculated. The experimental results are shown in Figure 16b.

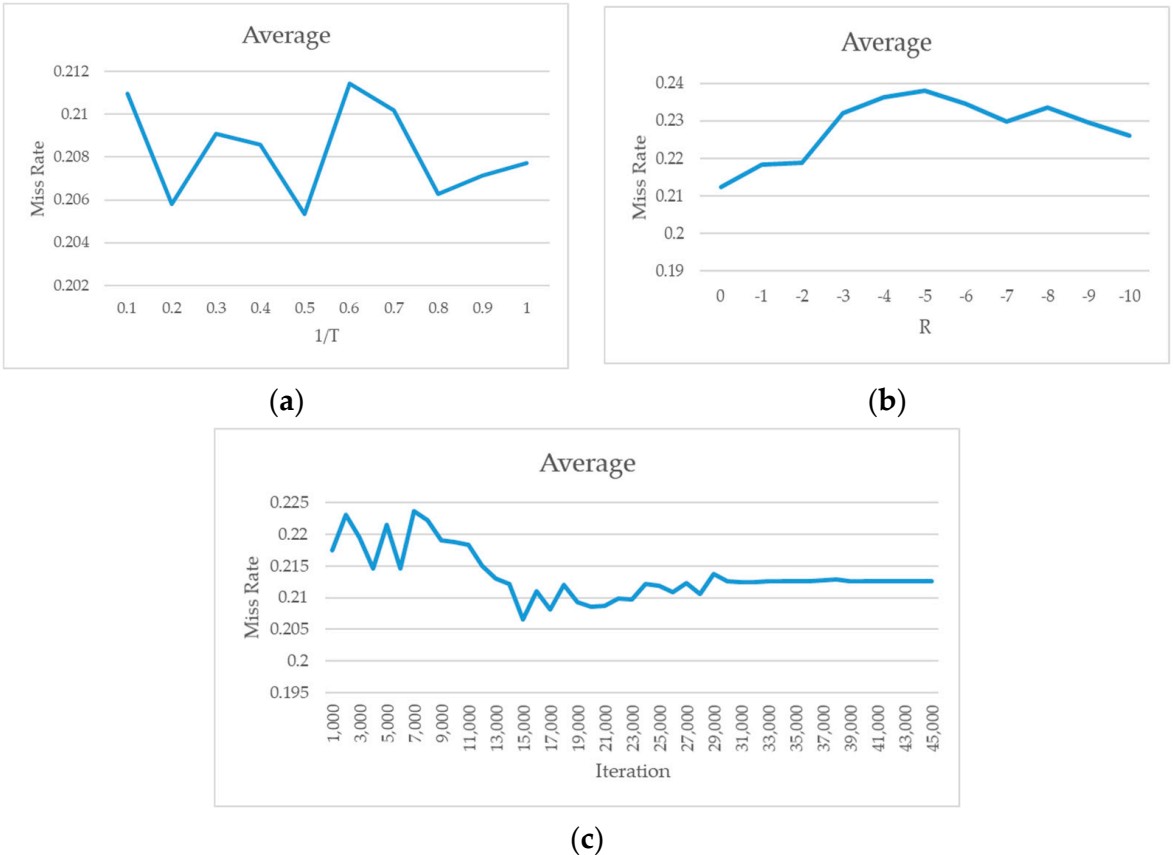

**Figure 16.** Miss rate curve obtained by (**a**) using different *1/T* fine-tuning parameters and (**b**) using different fine-tuning R parameters; (**c**) the number of iterations of fine-tuning.

In Figure 16b, we found that the average miss rate obtained using the evaluation had the lowest point: 0. When R = 0, the reserve rate of the non-domain-exclusive neurons was 50%. Thus, this value was too close to the reserve rate of the domain-exclusive neurons obtained in the T-parameter experiment (73.11%). It performed best on the PETS09 dataset and poorly on the other datasets. Therefore, it seems that R = −1 is the optimum value, with a reserve rate of 26.89%. This shows that the R parameters should not be too small and that non-domain-exclusive neurons should still have some fine-tuning space.

According to the aforementioned two experiments, the selection of the T and R parameters should be dominated by T and supplemented by R. The choice of T should refer to the maximum value of the NIS of each domain, such that the reserve rate is maintained at approximately 75%, which is $1/T = 0.5$ in this experiment. However, the choice of R should set the reserve rate of non-domain-exclusive neurons at approximately 25%, which is R = −1 in this experiment.

Finally, we discuss the iteration times of fine-tune training. Although fine-tuning has been widely used in many studies, there is no practical method for estimating the number of iterations that should be introduced in the fine-tuning stage. Therefore, in this experiment, we observed the performance of the model that was trained via domain-oriented fine-tuning for different iterations of each dataset on the basis of the investigation. In this experiment, 1000 iterations were used as units to increase the number of iterations to 45,000. A total of 45 models were trained. The base learning rate was 0.00002; for every 10,000 iterations, the learning rate was reduced by a factor of 0.1. Each trained model was evaluated using the Caltech test set, PETS09, and the DroneFJU datasets, and the average was calculated. The experimental results are shown in Figure 16c.

According to the experimental results shown in Figure 16c, we found that the miss rate of the model was minimal when the number of iterations was 15,000. If the training

was continued, the miss rate started to increase. The miss rate remained constant after approximately 30,000 iterations. This phenomenon may be because of the reduction of the learning to $2 \times 10^{-8}$, which may be too low, rendering the network unable to effectively continue learning. The optimal fine-tuned iteration time was set as 15,000.

On the basis of the aforementioned three experiments, we selected the best parameters to train the new model and evaluated its performance on the test set for the three pedestrian detection datasets. The chosen parameters were $1/T = 0.5$ and R $= -2$, and the iteration number was 15,000. R was $-2$ because a value of 0.5 was chosen for $1/T$ to maintain a reserve rate of 26.89%; $R/T = -1$ must be satisfied. Using the above formula, we obtain a new R $= -2$. The model was named for the best fine-tuning parameter as MDJL-ft-bv-cp*d*, and the results are listed in Table 5. Thus, the evaluation results in the gray background are the same as those in Table 4. The evaluation results were obtained by training the model using the single-domain dataset in the corresponding dataset.

**Table 5.** Average miss rate measured in each dataset using the model with the best fine-tuning parameter.

| Model | Average Miss Rate | | |
| :---: | :---: | :---: | :---: |
| | Caltech | PETS09 | DroneFJU |
| MS-CNN-c/-p/-d | 0.110483 | 0.544825 | 0.053416 |
| MDJL-cp*d* | 0.113062 | 0.487953 | 0.051723 |
| MDJL-ft-cp*d* | 0.107807 | 0.482616 | 0.046973 |
| MDJL-ft-bv-cp*d* | **0.103707** | **0.470637** | **0.045741** |

Table 5 summarizes that the model trained with the best parameters improved by approximately 1% in each domain, which demonstrates the importance of choosing the correct parameters for fine-tuning.

### 4.4. Comparison with State-of-the-Art Methods

The experiment was conducted in two parts. The first part compares the performances of the models trained using single-domain and multidomain datasets to highlight the associated advantages. The second part compares the proposed method with several state-of-the-art methods.

Table 6 indicates that in the past, when training a machine-learning method or deep learning network, we often chose a training dataset that was commonly used in the research community, such as the Caltech or INRIA datasets, to train the model. Given that the video data contained in these datasets generally have the same characteristics, such as the shooting environment, shooting angle, and equipment used, this type of dataset is called a single-domain dataset.

**Table 6.** Average miss rate compared with that of the state-of-the-art methods.

| Model | Classification | Training Dataset | Average Miss Rate |
| :---: | :---: | :---: | :---: |
| VJ [12] | AdaBoost | INRIA | 0.947337 |
| HOG [13] | Linear SVM | INRIA | 0.733415 |
| CCF + CF [19] | AdaBoost | Caltech | 0.173247 |
| Checkerboards+ [20] | AdaBoost | Caltech | 0.170986 |
| DeepParts [23] | DeepNet | Caltech | 0.11889 |
| MS-CNN-cp*d* [25] | DeepNet | cp*d* | 0.115921 |
| UDN+ [24] | DeepNet | Caltech | 0.115217 |
| MS-CNN [25] | DeepNet | Caltech | **0.110483** |
| MDJL-cp*d* | DeepNet | cp*d* | **0.113062** |
| MDJL-ft-cp*d* | DeepNet | cp*d* | **0.107807** |
| MDJL-ft-bv-cp*d* | DeepNet | cp*d* | **0.103707** |

Figure 17 shows the results for the cross-domain evaluation between several pedestrian detection models that were trained using a single-domain dataset and pedestrian detection models trained using a multidomain dataset. The dotted lines in the figure represent the results for a part of the single-domain dataset. We used HOG trained using the INRIA dataset as the traditional baseline method. In addition, we used the Caltech dataset, PETS09, and the DroneFJU datasets trained using MS-CNN as the baseline of the deep learning network. The solid lines in the figure represent the results for a part of the multidomain dataset. The MS-CNN and MDJL models were used, which were trained on the cp*d* dataset. For the evaluation, we used the Caltech, PETS09, and DroneFJU test sets.

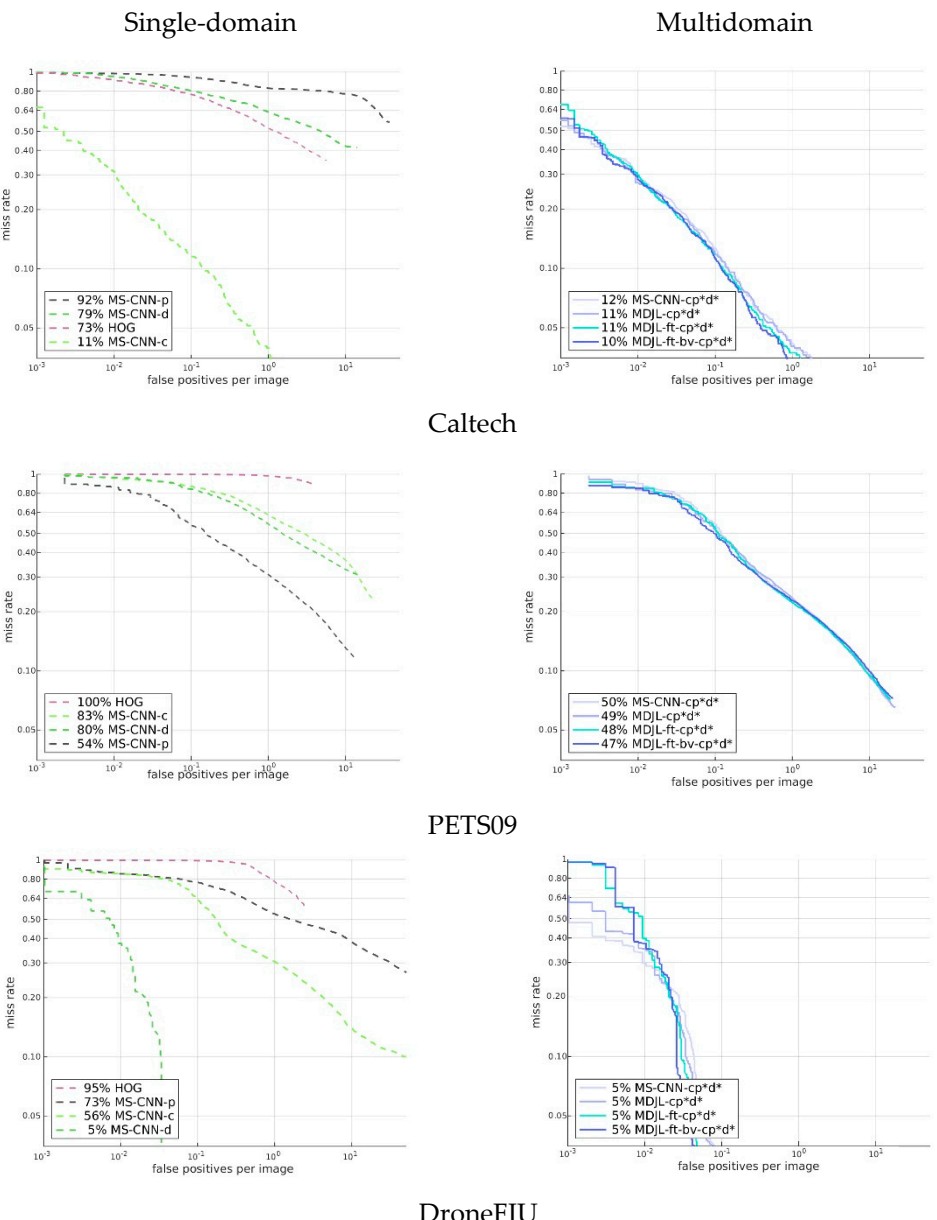

**Figure 17.** Single-domain dataset vs. multidomain dataset.

The results show that HOG was not observed in any dataset used in the evaluation in the training stage, so the effects on each dataset are not ideal. If we observe other models, we find that for the model trained using the single-domain dataset (dashed line), only the model trained with the training set of the same dataset performed better, and the rest were not ideal. For example, the average miss rate of MS-CNN for the Caltech dataset was 11%, but the average miss rates of MS-CNN-d and MS-CNN-p were as high

as 79% and 92%, respectively. This result shows that it is difficult to achieve cross-domain generalization using single-domain dataset training. Therefore, we believe it is essential to use multidomain dataset training.

Figure 17 shows that the model (solid line) trained with the multidomain dataset has comparable performance to the model (dotted line) trained using the corresponding single-domain dataset, regardless of the dataset. For example, in the evaluation results for the DroneFJU, the average miss rates of the three models, MS-CNN-cp*d*, MDJL-cp*d*, and MS-CNN-d, were similar. If we examine the evaluation results of PETS09, the average miss rates of MS-CNN-cp*d* and MDJL-cp*d* using the multidomain dataset are 4% and 5% lower than that of MS-CNN-p, respectively. These results show not only that using multidomain dataset training allows the model to have cross-domain generalization capabilities but also that domains can occasionally have complementary effects.

Next, we compared the proposed method with current state-of-the-art approaches. Results were obtained for MS-CNN, MS-CNN-cp*d*, and MDJL-cp*d*. The data for the other algorithms were obtained by using the data on the Caltech pedestrian detection benchmark website. The classification methods used for each method and the dataset used for training are listed in Table 6.

Receiver-operating characteristic (ROC) analysis is a valuable tool for evaluating the performance of diagnostic tests and the accuracy of a statistical model's outcomes. We compared MDJL with state-of-the-art methods and drew the ROC curve shown in Figure 18. From Figure 18 and Table 6, it is evident that although MDJL used a multidomain dataset to train the network, it can still maintain the performance of the state-of-the-art pedestrian detection benchmark and can beat MS-CNN after fine-tuning.

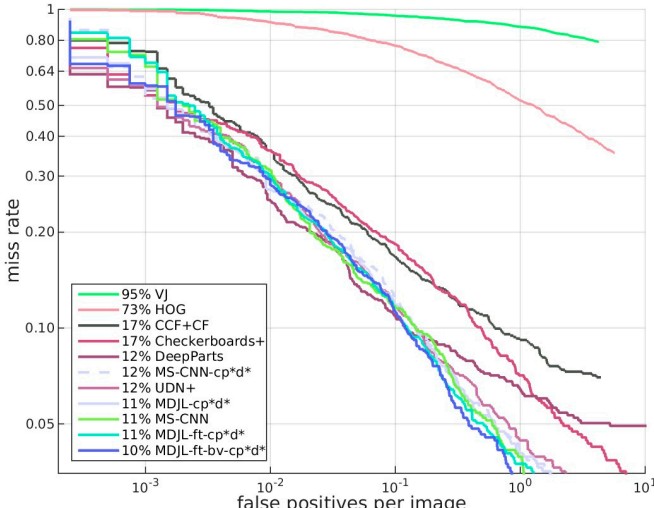

**Figure 18.** ROC line chart compared with state-of-the-art methods.

### 4.5. Pedestrian Tracking

In this experiment, we considered pedestrians as the primary tracking objects. In the experiment, we used the four videos from the MOT Challenge 2015 as the training and test data. The training part of the MDP tracker used S2L1 in PETS09 and TUD-Stadtmitte [38] (Table 7), whereas the experimental evaluation used S2L2 in PETS09 and TUD-Crossing as test videos.

**Table 7.** Experimental dataset.

|  | FPS | Resolution | Frame | Tracks |
|---|---|---|---|---|
| PETS09-S2L1 | 7 | 768 × 576 | 795 | 19 |
| PETS09-S2L2 | 7 | 768 × 576 | 436 | 42 |
| TUD-Crossing | 25 | 640 × 480 | 201 | 13 |
| TUD-Stadtmitte | 25 | 640 × 480 | 179 | 10 |

In this experiment, two videos were selected in PETS09, namely S2L1 and S2L2, for training and testing, respectively. Both videos were acquired in an open outdoor environment, and the shooting angle was from the perspective of a fixed surveillance camera with a downward projection. In the PETS09 video set, several crowded groups of pedestrians walk freely, and there are numerous occlusions between the pedestrians, which leads to difficulty in detection or tracking. Figure 19a is a screenshot of a region of the pedestrian scene in the video set.

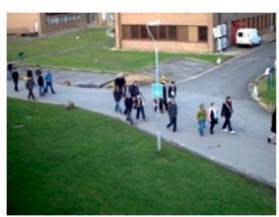 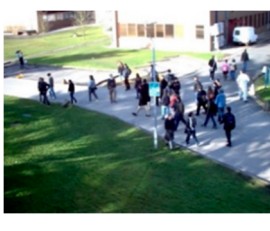 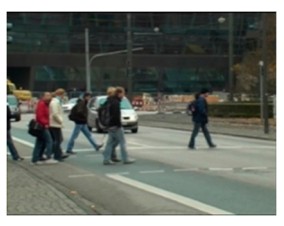 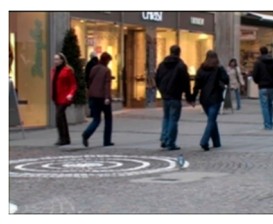

(**a**)                                               (**b**)

**Figure 19.** (**a**) PETS09 dataset and (**b**) outdoor street scene.

The TUD-Stadtmitte and TUD-Crossing are outdoor street scenes, as shown in Figure 19b. The difference compared with PETS09 is that the shooting angle is fixed, and image acquisition is performed from a forward projection. Many pedestrians are observed walking along the street in the video. However, because of the forward projection, the occlusion between pedestrians is more significant compared with the surveillance view with a downward projection, which makes it more difficult to track the trajectory and associate the bounding box.

In the performance evaluation, multiple evaluation criteria [39] were used for the MOT benchmark [40], including multiobject tracking accuracy (MOTA), multiobject tracking precision (MOTP), ID switch, false positive (FP), false negative (FN), mostly tracked (MT), and mostly lost (ML).

FP denotes the total number of marking errors in the bounding box, and FN is the total number of unmarked ground truths. The ID switch is an important criterion for evaluating multiobject tracking, because the primary goal of this process is to obtain the object ID and monitor the trajectory. The definition of MOTA is given in Equation (11). MOTA is used primarily to evaluate the tracking accuracy of the object's trajectory. The referenced error-rate scores include the FN, FP, and ID switches, which are useful indicators. Compared with MOTA, MOTP does not consider the erroneous result of the ID switch but evaluates the tracker's accuracy in estimating the position and size of the object. The MOTP is calculated using Equation (12). First, the total error between the actual and estimated positions of all targets in the frames was counted and averaged. $c_t$ is the number of objects in the ground truth of the $t$-th frame, and $d_t^i$ is the overlapping rate of the bounding box of the $i$-th target in the $t$ frames. Next, MT and ML were used to evaluate the integrity of the trajectory; however, the ID switch was not considered. If the trajectory time of an object was successfully tracked for more than 80%, then the trajectory was classified as MT; if it was only successfully tracked for less than 20%, then the trajectory was classified as ML.

$$OTA = 1 - \frac{\sum_{t=1}^{N_{frame}} (FN_t + FP_t + IDSW_t)}{\sum_{t=1}^{N_{frame}} GT_t} \tag{11}$$

$$MOTP = \frac{\sum_{i,t} d_t^i}{\sum_t c_t} \tag{12}$$

In this experiment, we determined the system's best parameters by observing the influence of each parameter on the final tracking result. We adjusted the parameters to filter the pedestrian detection results: confidence threshold, nonmaximum suppression

(NMS) threshold, and the bounding box threshold (parameter for pedestrian tracking). We used PEST09-S2L2 and TUD-Crossing as test videos and averaged the results.

The first parameter of pedestrian detection is the confidence threshold, which is used primarily to filter the bounding box obtained via pedestrian detection. This parameter affects the number of FP and FN. When the threshold is high, the detection results have a low FP, but it is relatively easy to obtain a higher FN. The effect of this threshold adjustment on the tracking results is shown in Figure 20a. The results show that this parameter has little impact on MOTP. However, when the threshold is too high or too small, MOTA and MT begin to decline, whereas ML increases. It is speculated that when FN is higher than a threshold value, the tracker is unable to associate with the target, and the trajectory is broken. Thus, the MT decreases, and the ML increases. The level of MOTA depends on the number of FN and FP. It was determined that the optimal parameter was 0.6, which resulted in the highest MOTA.

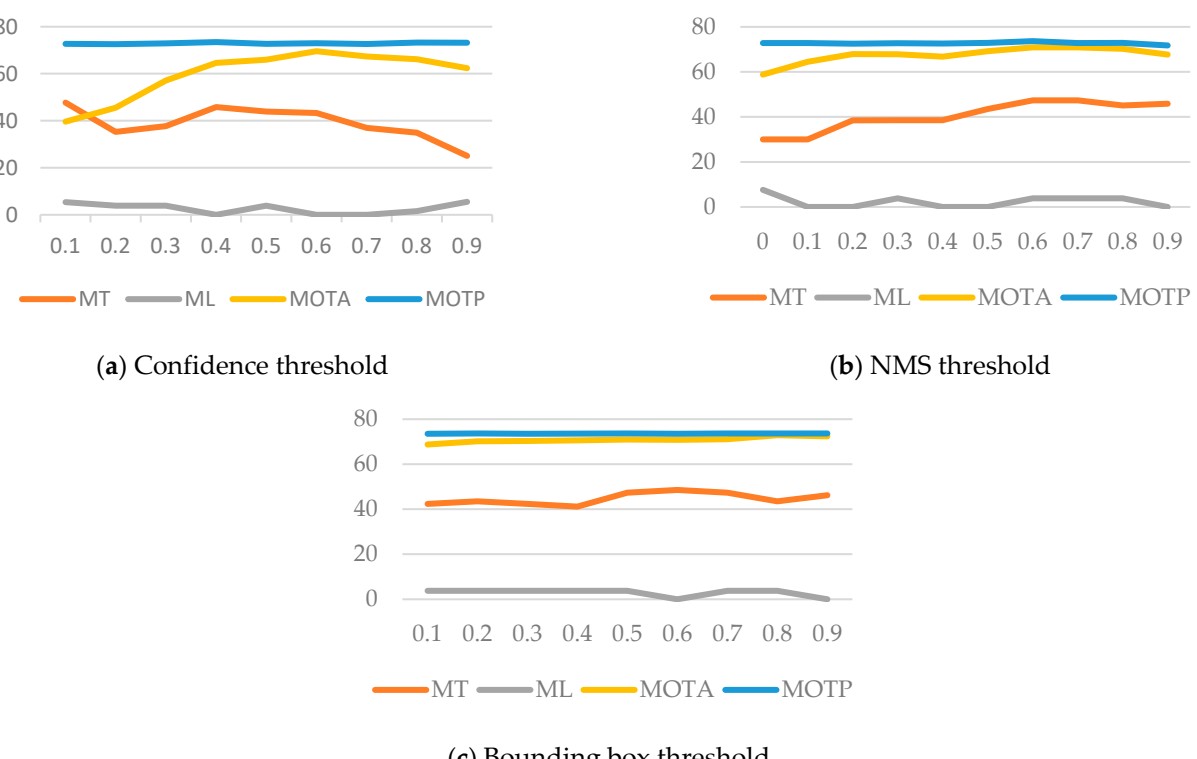

**Figure 20.** Influence of the adjustment of pedestrian detection parameters.

The second parameter for pedestrian detection was the NMS threshold. This parameter affects the accuracy and the quantity of the bounding box positions. The effect of this threshold adjustment on the tracking results is shown in Figure 20b. The results show that when the threshold is high, MOTP decreases slightly, and MT increases simultaneously. It is speculated that when the threshold is high, some bounding boxes have too many redundant detection results because of the inability to merge, which increases the number of FPs and decreases the value of MOTP. Conversely, when the threshold is lower, most bounding boxes obtain the best detection position. However, if multiple objects are too close, the bounding boxes of various objects may merge, thereby increasing the number of FN. Therefore, the MOTP will be higher, but the MT will be lower. Therefore, the optimum parameter was determined to be 0.6, which maintained the highest MOTA and MOTP.

Finally, we adjusted the pedestrian tracking parameter bounding box threshold. When the overlap rate between the pedestrian position predicted by the tracker and the bounding box was higher than this threshold, the algorithm associated the bounding box with the tracker and then continuously performed tracking. This parameter directly affects the tracking accuracy. The effect of this threshold adjustment on the tracking results is

shown in Figure 20c. The results show that MT increases to a specific level when the threshold increases and then subsequently decreases. It is speculated that the tracker is more prone to associate errors when the threshold is low. Therefore, both MT and MOTA were relatively low. The tracker can associate the correct bounding box more accurately when the threshold increases. However, when the threshold is high, the tracker has a low error tolerance because of the threshold; therefore, both MT and MOTA decrease. Thus, the optimal parameter was chosen to be 0.8, which maintained the highest MOTA.

After adjusting the optimal parameters, the system was referred to as MDP_MDJL. Next, we compared our system with the current state-of-the-art methods in several experiments. The test videos used were PETS09-S2L2 and TUD-Crossing. Our method was compared with the six most advanced tracking techniques using the MOT benchmark, including two offline tracking methods, TSML_CDE [41] and DMT [42], and four online tracking methods, CDA_DDAL [43], AMIR [44], APRCNN [45], and MDP [33].

The results for PETS09-S2L2 are shown in Figure 21, which indicates that the proposed approach has the best MOTA and MOTP compared with the other methods. Compared with MDP, MOTA and MOTP increased by 18.8% and 5.3%, respectively. The result for the TUD-Crossing is shown in Figure 22; our approach has the best MOTA compared with the other methods. Compared with MDP, MOTA and MOTP increased by 9.7% and 4.7%, respectively.

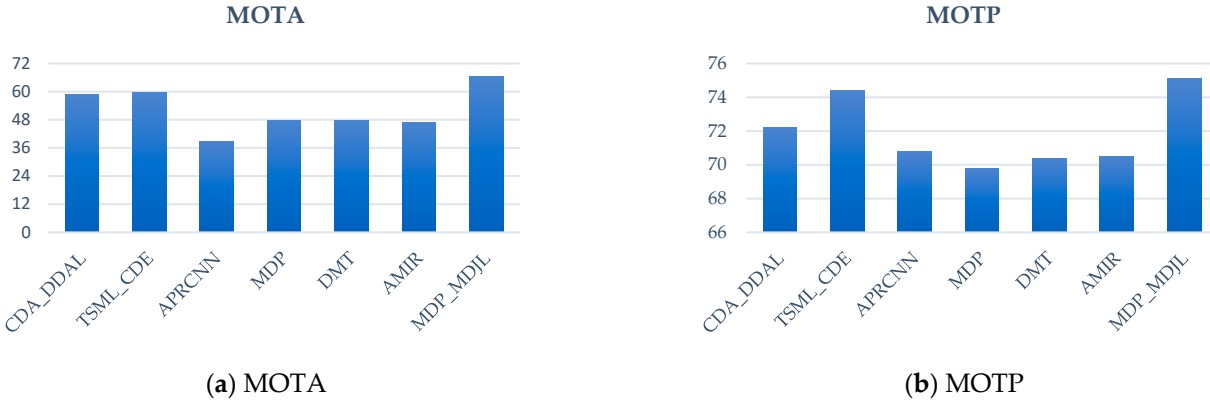

(**a**) MOTA         (**b**) MOTP

**Figure 21.** PETS09-S2L2 analysis chart.

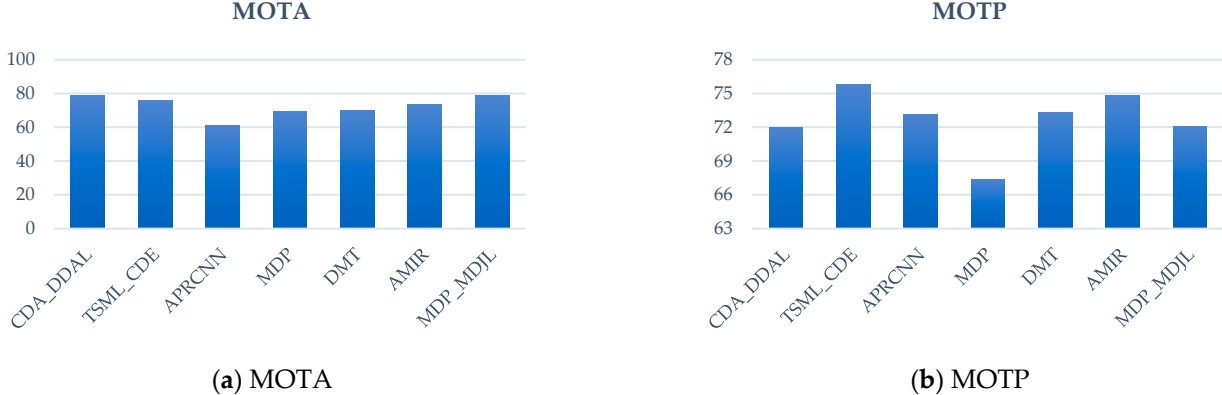

(**a**) MOTA         (**b**) MOTP

**Figure 22.** TUD-Crossing analysis chart.

Overall, the proposed method performed best on the MOTA indicator in the two experimental video scenarios. This was primarily because of the high accuracy of MDJL in the case of multiple domains. In addition, optimizing the parameters allowed the detector to obtain the most accurate position, thus yielding the best performance.

## 5. Conclusions

This paper discussed the common problem of appropriate datasets often being unavailable in the implementation of machine learning to pedestrian detection and tracking. To address this issue, we proposed the MDJL method. This approach can effectively address the phenomenon in which the network performance in each domain is not ideal when multiple datasets are used for training. Moreover, it can improve the network's performance in similar domains. In addition, the MJDL model was more generalizable. When the number of samples in each dataset differed significantly, to prevent a large dataset from dominating the network and affecting the performance of the other domains, we used data augmentation to balance the difference in the number of samples between the domains. In addition, we also combined the MDJL and MDP trackers to create a multiobject tracking system that is suitable for flying drones. We used the characteristics of the MDJL model to enable the tracking system to effectively cope with a photography platform with a high degree of spatial freedom, such as an aerial camera. The experimental results also showed that MDJL can cope with different scenarios and significantly improve the systems' tracking performance.

When samples are expanded using data augmentation, it may be useful to consider referring to the characteristics of the original dataset; it may yield different training results. In addition, the drone's camera mobility and high shooting angle are not available in fixed-position surveillance and driving recorders. An inappropriate viewing angle leads to more misjudgments. Improvement in the pedestrian tracking recognition rate for horizontal viewing angles should be investigated in the future.

**Author Contributions:** Conceptualization, Y.-K.W. and J.G.; methodology, Y.-K.W. and J.G.; software, T.-M.P. and J.G.; validation, T.-M.P. and J.G.; formal analysis, Y.-K.W. and J.G.; data curation, J.G.; writing—original draft preparation, J.G.; writing—review and editing, T.-M.P. and Y.-K.W.; supervision, Y.-K.W. All authors have read and agreed to the published version of the manuscript.

**Funding:** This research received no external funding.

**Conflicts of Interest:** The authors declare no conflict of interest.

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
