# Peer review of "Multidomain Joint Learning of Pedestrian Detection for Application to Quadrotors"

_drones, doi:10.3390/drones6120430_

Round 1

Reviewer 1 Report

Overall paper is good and well-written.

1. Try to portray the full study in short form in the abstract.

2. Figures 2 and 14 are hazy and acceptable. Redraw it.

3. Authors conducted any ablation study to build and select the layers of the fine-tuned network? If not, I highly recommend it and the ablation results.

4. Do you have conducted any statistical testing of your model and outcomes? If not, please employ at least one statistical test.

5. Authors have provided lots of equations, please provide the purposes and importance of the equations of proper descriptions.

6. Most of the citations are old, try to use recent studies (2019-2022).

7. Grammatical correction is needed.

Author Response

Dear Reviewer:

Reviewer 2 Report

The authors show us a new MDJL-based pedestrian detection method with application to the UAV domain. I think this method is relatively new from the perspective of engineering applications and has the value of inclusion. However, the authors need to make some corrections to the article before proceeding to the next step of the paper processing.

(1) The writing in the related work section is disorganized and lacks logic. The explanatory text in the dissertation template appears in the last two paragraphs of the related work. The author did not proofread the manuscript carefully; please re-correct all relevant sections.

(2) The font in the picture is different from the picture, for example, in the section of Figure 2. Please check all similar cases.

(3) Please standardize the formatting of tables in the article.

Author Response

Dear Reviewer:

Reviewer 3 Report

This work propose a novel multi-domain joint learning (MDJL) pedestrian detection method, by exploiting data from various datasets from multiple domains, using neural networks. The work propose a training method using domain-guided dropout to use network domain information to enhance the performance of the network when learning. The paper presents a good bibliographic review, with high impact citations.

Although the subject matter is very interesting, in my opinion, the presentation of the work is not very careful. Here are some examples of this:

-        Lines 16-17. The verb propose is used twice in the same sentence in a wrong way: “we propose a method”, and “a method is proposed”.

-        Line 93, 236, 278: It refers to the work as a report. It is not a report, it is a paper, or research work.

-        Lines 189-193. It should not appear, what is said should be applied.

-        Lines 194-196. It should not appear, what is said should be applied.

-        Lines 479-486 y Lines 488-501. Explain the same concept repeatedly

-        Line 667. Refers to a figure, and does not indicate the number

-        In point 4. Experimental Results, when describing the experimental data, the data forming the different domains are not described in detail.

On the other hand, the literature review does not cover all the methodology used, with few high impact citations. A better literature review should be carried out, in journals of higher impact

Author Response

Dear Reviewer:

Round 2

Reviewer 3 Report

  • The paper is ready for publication